# Disclosing disabilities: Barriers for medical school applicants

Sarah Leah Solomon[1]*, Aitan E. Magence[2], Harry Haran[3], Fern R. Juster[4], Kristina H. Petersen[5]

1 Psychiatry Department, Dartmouth-Hitchcock Medical Center, Lebanon, New Hampshire, United States of America, 2 Pediatrics Department, Westchester Medical Center, Valhalla, New York, United States of America, 3 School of Medicine, New York Medical College, Valhalla, New York, United States of America, 4 Admissions Department - Emeritus, New York Medical College, Valhalla, New York, United States of America, 5 Biochemistry and Molecular Biophysics Department, Washington University School of Medicine, St. Louis, Missouri, United States of America

☯ These authors contributed equally to this work.
* sarah.l.solomon@hitchcock.org

## Abstract

This study investigated the availability of accommodation request procedures for medical school admissions interviews and the accessibility of Technical Standards (TS) for candidates with disabilities (CWD) across Liaison Committee on Medical Education (LCME)-accredited institutions in the United States and Canada. Utilizing a cross-sectional study methodology, surveys were distributed to Deans of Admissions and Disabilities Resource Professionals (DRP) at all LCME-accredited US and Canadian MD programs. Surveys gathered data about interview formats, interview accommodation procedures, and TS accessibility during the 2018–2019 academic year. We received responses from 71 institutions (41.3%), with 56.3% survey completion rate (n = 40). Among respondents, interview formats varied: 26.8% (n = 19) Multiple Mini Interview, 32.4% (n = 23) Traditional Interviews, and 16.9% (n = 12) hybrid. 38% (n = 27) of respondents informed CWD of accommodation procedures before interviews. Ten institutions (14.1%) indicated they had updated their procedure since the 2018–2019 academic year, which demonstrated better overall outcomes. Statistical analyses showed significant differences between institutions with/without updated procedures in the total number of applicants who requested accommodations, were granted interviews, provided interview day accommodation, offered admission, and matriculated (p = 0.005). In 66.7% (n = 18) of respondent institutions, admissions staff were aware of initial interview accommodation requests and 44.4% (n = 12) involved admissions staff when communicating accommodation plans. Among 27 schools, 55.6% (n = 15) required no documentation to support the CWD's need for accommodation; the rest required a form, clinician's letter, past proof, or other methods. 56.3% (n = 40) responded questions about TS and confirmed posting them on their website, with 77.5% (n = 31) on their admissions webpage. 77.5% (n = 31) also reported

**Data availability statement:** All relevant data are within the paper and its Supporting Information files.

**Funding:** The author(s) received no specific funding for this work.

**Competing interests:** The authors have declared that no competing interests exist.

including language in the TS that direct CWD to the institution's disability office. This study reveals communication deficiencies about accommodations and TS requirements during the admissions process. Recommendations to enhance accessibility include informing candidates early about accommodation procedures and TS, and utilizing DRPs as CWD's primary accommodation contact.

## Introduction

One in four persons in the US identify as having a disability, in comparison to 2–10% of physicians [1,2]. Medical school candidates with disabilities (CWD)—including those with physical, sensory, psychological, learning, or chronic health conditions—encounter numerous barriers when applying for admission. Many obstacles arise before they submit an application, including challenges obtaining accommodations on the Medical College Admissions Test (MCAT), the Situational Judgement Test (SJT), and/or interview formats that assess responses to hypothetical scenarios (often referred to as situational or behavioral interviews). While numerous barriers exist to overcome these pre-application hurdles, the authors limit the scope of this paper to the interview process.

### Interview format

The Multiple Mini Interview (MMI) format was created in 2001 as an alternative to the traditional interview (TI); studies have shown that the MMI format decreases bias and increases reliability through a candidate's engagement with multiple interviewers and the use of an established grading rubric [3–5]. While this is encouraging, the MMI can present specific barriers for CWD. For example, CWD may be entitled to extended time to read prompts or breaks between stations to accommodate their disabilities. Traditional interviews may not present these exact challenges, but they are not without their own barriers. For instance, TI's may require adjustment to the interview structure or timing. Additionally, virtual interviews—whether MMI or TI—can introduce new accessibility issues, including platform or communication difficulties. As illustrated, regardless of the interview format or delivery method, accommodations are often necessary to ensure equitable access for all candidates. CWD who engage in the medical school interview process must weigh the perceived benefits and risks before disclosing their disability status to request accommodations. In these cases, candidates may feel compelled *not* to disclose their disability status and forgo accommodations rather than risk bias in their admission decision. The decision is often accompanied by significant emotional turmoil, including anxiety, fear of judgment, and internal conflict. Candidates may struggle between protecting their privacy and advocating for their needs. Ideally, a process should direct applicants to a neutral, knowledgeable individual or office, such as a Disability Resource Provider (DRP), as is best practice for matriculated students [6–8]. This approach can help mitigate emotional distress by assuring a certain level of confidentiality. Without a confidential process for requesting necessary interview accommodations, candidates may need

to disclose their disability to the admissions office which presents a conflict of interest and may unintentionally discourage CWD from disclosing their need for accommodation. Notably, this study asked institutions to comment on their AY 2018–2019 procedures when most institutions were still holding in-person interviews, and thus CWD likely had a variety of accommodation requirements.

## Technical standards

Technical Standards (TS) describe the essential functions that students must demonstrate to meet the medical education requirements at a given institution for entrance, promotion, and graduation [9]. TS may pose barriers for CWD if they do not explicitly allow for accommodations [10–13]. When TS are written with an inclusive approach, they acknowledge that students may use accommodations to meet the standards [10,11,14–16]. Unfortunately, many US medical schools do not include this acknowledgment, a practice that may discourage CWD from applying [10].

Significant barriers to admission exist for otherwise qualified medical school CWD [10,14–18] that perpetuate their exclusion, countering many medical schools' mission statements and broader calls for increased diversity and inclusion [19]. To date, no studies have identified barriers associated with obtaining accommodations during the admissions process; this information may provide insight into proactive measures that can be taken to bridge the discrepancy between the proportions of physicians with disabilities and those with disabilities in the general population. The AAMC first piloted disability-related questions in its 2018–2019 Graduation Questionnaire (GQ), noting that only 2.3% disclosed disabilities [20]. Comparatively, 11.6% disclosed disabilities on the 2024 GQ [21]. While this increase over five years is encouraging, a multi-pronged approach is necessary to identify barriers to admission that may prevent otherwise qualified CWD from applying, seeking interview accommodations, or being admitted.

## Current study

Although research has been conducted on the barriers that CWD encounter due to non-inclusive TS, more investigation is necessary to fully characterize and measure the impact of barriers to the medical school interview accommodation process. This includes examining the point at which candidates are directed to review the TS and if/when candidates are informed of a process to request interview accommodations. Our study aims to evaluate the procedures in place at Liaison Committee on Medical Education (LCME)-accredited allopathic medical schools throughout the US and Canada regarding interview accommodations and directing students to review the TS. We hope to better understand current practices and make recommendations to enhance accessibility and inclusivity.

## Methods

A multi-institutional, cross-sectional survey study was conducted to examine medical school admissions interview accommodation procedures and the timing of referring applicants to review each program's TS (S1 and S2 Appendix). The survey was developed by the study authors based on a review of relevant admissions processes and established survey design practices. It was piloted with admissions staff who were knowledgeable about interview procedures but not eligible to participate in the study. Feedback was used to revise item wording, improve clarity, and confirm the relevance of content. The finalized survey included a combination of multiple-choice, Likert-type, and open-ended questions organized into four sections: (1) admissions cycle 2018–2019 data on interview formats and accommodations; (2) 2018–2019 procedures for requesting interview accommodations, including documentation requirements and public availability; (3) any changes to these procedures since the 2018–2019 cycle; and (4) the online accessibility of the TS, including whether or not contact information was specified for each institution's Office of Disability or Accessibility.

The web-based survey was administered using Qualtrics and distributed via email to the Deans of Admissions at 155 U.S.-based and 17 Canadian-based LCME-accredited MD programs (S1 Appendix). To support institutional participation,

the survey was also sent to each school's disability resource professional (DRP). Contact information was identified through publicly accessible institutional websites. The survey was open from June 9, 2021, through October 2, 2021. Four reminder emails were sent to encourage participation. The survey focused on data from the 2018–2019 application cycle to avoid distortions caused by the COVID-19 pandemic, which led to significant mid-cycle shifts in interview formats during 2019–2020. Responses were anonymous, and no incentives were offered for participation. The study was reviewed and deemed exempt by the New York Medical College Institutional Review Board (IRB# 14478).

### Data analysis

Initial descriptive analysis and data cleaning were performed using Microsoft Excel. Incomplete survey responses were included in descriptive summaries but analyzed separately in inferential tests where appropriate. All statistical analyses were conducted using IBM SPSS Statistics for Windows, Version 26. Descriptive statistics were calculated for each survey item, including counts, percentages, means, and standard deviations.

Inferential analysis included a one-way analysis of variance (ANOVA) to evaluate differences in outcomes based on whether institutions reported changing or amending their procedure for handling interview day accommodation requests since the 2018–2019 application cycle. The independent variable for this analysis was institutional procedure status ("Yes, new procedure," "Yes, amended," or "No change"). Dependent variables included: (1) the number of applicants who requested interview day accommodations, (2) the number who were granted accommodations, (3) the number subsequently granted admission, and (4) the number who matriculated. For ANOVA results that were statistically significant, a Tukey honest significant difference (HSD) post hoc test was used to identify pairwise differences between groups.

Independent samples t-tests were used to assess whether the presence of technical standards on the admissions website or inclusion of a statement directing applicants to the disability resource office was associated with the number of applicants granted accommodations, admitted, matriculated, or registered with the disability office. Kruskal-Wallis H tests were used to examine associations between outcome variables and the identity of the person or office who served as the initial contact or final decision-maker for interview day accommodations. Additionally, a chi-square test was used to assess whether the identity of the initial contact person was associated with whether any applicants granted accommodations were ultimately admitted. For each question analyzed, we included the number of respondents who did not answer the question to ensure accurate reporting.

### Results

A total of 172 medical schools were invited to participate in the survey, yielding a response rate of 41.3% (n = 71). Among these, 62 were U.S.-based institutions and 9 were Canadian. Of the total responses, 56.3% (n = 40) completed all parts of the survey. Completion rates were 53.2% for U.S. institutions (n = 33) and 77.8% for Canadian institutions (n = 7)..

### Interview format

Institutions were asked to identify the format of their admissions interviews. Of the respondents, 26.8% (n = 19) reported using multiple mini-interviews (MMIs), 32.4% (n = 23) used traditional interviews, 16.9% (n = 12) used a hybrid approach, and 23.9% (n = 17) did not specify an interview format. Canadian institutions were more likely to report using MMIs (77.8%) compared to U.S. institutions (19.4%).

### Accommodation communication

Respondents were asked whether their institution provided invited applicants with a clear process for requesting interview day accommodations. Across all 71 institutions, 27 (38.0%) reported providing this information, 27 (38.0%) reported not providing it, and 17 (23.9%) chose not to answer the question. Communication of procedures was more common among Canadian institutions (8/8, 100%) than U.S. institutions (19/46, 41.3%) among those that responded.

## Institutional procedure changes and associated outcomes

Institutions were asked whether they had added, changed, or amended their interview day accommodation procedure since the 2018–2019 admissions cycle. Ten institutions (14.1%) reported that they had either implemented a new procedure or amended an existing one.

A one-way ANOVA was conducted to assess whether changes to institutional procedure were associated with differences in applicant outcomes. Significant differences were found between groups for each outcome: number of requests ($F_{(2, 29)} = 6.46$, $p = .005$); number granted accommodations ($F_{(2, 30)} = 6.78$, $p = .004$); number admitted ($F_{(2, 23)} = 6.89$, $p = .005$); and number matriculated ($F_{(2, 23)} = 10.84$, $p < .001$).

Post hoc Tukey HSD tests revealed that institutions with amended procedures had significantly higher average values for each outcome compared to those with no reported change. Specifically, the mean differences between these two groups were 1.35 for applicants granted accommodations, 2.62 for those admitted, and 1.33 for those who matriculated (all $p < .01$). A post hoc power analysis assuming an effect size of 0.4, alpha of 0.05, and three comparison groups indicated adequate statistical power (critical $F = 3.34$).

## Initial contact person and decision-maker

To explore whether administrative roles influenced outcomes, institutions were asked to identify who served as the initial point of contact for accommodation requests and who made the final determination. The majority of respondents did not provide a valid answer to these two questions (44 out of 71; 62%). Among the 27 valid responses, the most common initial point of contact for applicants requesting interview day accommodations was the Office of Admissions (n = 16, 59.3%), followed by the DRP (n = 7, 25.9%), Dean of Admissions (n = 2, 7.4%), OSA (n = 1, 3.7%), and OUME (n = 1, 3.7%). Canadian institutions (n = 7 responses) most frequently listed the Office of Admissions (n = 4, 57.1%) and the Disability Resource Professional (n = 2, 28.6%) as the primary contact for interview accommodations, with one school (n = 1, 14.3%) citing OUME. In contrast, American institutions (n = 20 responses) predominantly identified the Office of Admissions (n = 12, 60.0%) and DRP (n = 5, 25.0%), with a few referencing the Dean of Admissions (n = 2, 10.0%) or OSA (n = 1, 5.0%).

When asked to identify the office responsible for making the final determination and communicating with applicants, responses included the Dean of Admissions (n = 8, 29.6%), DRP (n = 6, 22.2%), other offices (n = 7, 25.9%), Office of Admissions staff (n = 4, 14.8%), and OSA (n = 2, 7.4%). Canadian schools (n = 7) most frequently reported that final accommodation decisions were made by "Other" offices (n = 3, 42.9%) or the DRP (n = 2, 28.6%), whereas American schools (n = 15) most commonly indicated the Dean of Admissions (n = 7, 46.7%) or the DRP (n = 4, 26.7%) as the final decision-maker.

Kruskal-Wallis H tests were conducted to examine differences in outcomes based on administrative roles within the interview accommodation process. No significant differences were found in the number of applicants granted accommodations, admitted, matriculated, or registered with the disability office (all $p > 0.26$), suggesting that administrative structure alone was not associated with accommodation outcomes.

Additionally, a chi-square test was conducted to evaluate the association between the initial point of contact for accommodation requests and whether any applicants granted accommodations were admitted. The association was not statistically significant, $\chi^2(2, N = 13) = 0.48$, $p = 0.786$, indicating that whoever assumed the specific contact role was not related to admissions outcomes for accommodated applicants.

## Documentation

Twenty-seven medical schools responded to a "select all that apply" question regarding documentation requirements for interview day accommodations in the 2018–2019 application cycle. Among them, 5 schools (18.5%) required a unique application form specifically for interview accommodations, 5 (18.5%) required at least one letter from a clinician, and 4 (14.8%) required proof of past accommodations (e.g., from undergraduate studies or standardized testing). Notably, 15

schools (55.6%) reported that no application form or documentation was required. Additionally, 7 schools (25.9%) selected "Other" and provided open-text responses.

## Technical standards

Institutions were also asked whether their TS were posted on their admissions website and whether the standards included language directing applicants to disability resources. Of the 40 (56.3%) schools that chose to respond, 31 (77.5%) had TS publicly posted on their admissions website. Among the 40 school respondents, 31 (77.5%) included a statement within the TS directing students to the office for disability resources, while 9 (22.5%) did not. Additionally, 77.8% (7/9) of Canadian institutions reported posting their TS on their website, and 100% (7/7) on their school's admissions website page, compared to US institutions in which only 53.2% (33/62) reported posting this on their website and 87.9% (29/33) on their school's admissions website page. Independent samples t-tests were used to assess whether these elements were associated with the number of applicants granted accommodations, admitted, matriculated, or registered with the disability office. No statistically significant differences were found for any outcome when comparing institutions: with or without visible TS on their admissions website (all $p > .44$), or TS with or without a disability office referral (all $p > .47$).

## Discussion

This is the first study to assess the medical school admissions interview accommodation request and TS referral procedures at LCME-accredited institutions in the US and Canada. Overall, data analyses reveal a lack of institutional process to ensure CWD are provided with interview accommodation request procedures and are referred early to TS. Although our response rate was not as strong as we would have liked, these results highlight significant oversight in accommodating the needs of CWD that warrant further exploration.

### Interview format

To contextualize potential candidate accommodation needs at various institutions, we first must discuss the variety of interview formats utilized. Despite the known advantages of using the MMI format to reduce interview bias, our study found that nearly half (49.3%) of the responding medical institutions still utilize some form of the TI, with almost a third (32.4%) exclusively employing the TI format. A comparison between US and Canadian institutions revealed that US respondents (19.4%) utilized the pure MMI format at a much lower rate compared to Canadian respondents (77.8%). Notably, while shown to reduce bias in general, MMI studies have not specifically explored bias among CWD. However, if a candidate who requires accommodations does not request them, and consequently does not perform to their best ability, this may call the validity of the interview, regardless of format, into question. Furthermore, since data were collected for the 2018–2019 admission cycle, many schools were holding in-person interviews, a process that inherently lends itself to more accommodation needs than remote interviews (e.g., additional break time or wheelchair accessibility). Hence, it is likely that many candidates had accommodation needs; however, our results suggest that the process required to request interview-day accommodations could be better elucidated and communicated to candidates.

### Accommodation request procedures

Our results raise concerns about the lack of communication with US medical school candidates about interview accommodation request procedures. Only 38% of respondent institutions confirmed notifying students of these procedures, while 38% chose not to answer the question. Our results are notable for the number of non-responses to several key questions. While survey fatigue may have played a role, it is unlikely that institutions lacked the necessary information for a straightforward 'yes' or 'no' response. The absence of institutional data tracking alone may not fully explain unanswered questions. Other possible explanations include respondents not knowing the answer or being hesitant to admit that their

institution lacked certain procedures. However, the former seems unlikely, considering each institution's DRP involvement in providing necessary information per the study's protocol. Acknowledging such oversights may be frustrating, but recognizing shortfalls is the first step toward improvement. Notably, among Canadian medical institutions, we saw contrasting results: 78% confirmed providing interview accommodation procedure information to candidates.

### Institutional procedure changes and associated outcomes

Ten institutions reported revising their designated contact person for interview accommodation requests since the AY 2018–2019. Encouragingly, analyses showed that institutions that implemented changes to their procedures had higher numbers of applicants requesting disability-related interview accommodations, granting those accommodations, and subsequently matriculating. This suggests that institutions aware of disability considerations were already providing more interview accommodations and admitting more CWD before altering their interview accommodation procedures.

### Initial contact person and decision-maker

When evaluating the accessibility of medical school interview accommodation procedures, identifying the point of contact for requests is crucial to avoid conflicts of interest. We asked respondents to identify the initial contact person and the individual communicating the accommodations plan to the applicant. Unfortunately, almost two-thirds (62%) of responding institutions did not provide an answer to either of these questions. Among matriculated students, the recommended procedure is to direct applicants to a neutral, knowledgeable individual or office, ideally the DRP [6–8]. This same principle applies to CWD seeking admission. Less ideal options may include the OSA or the OUME, which lack expertise in disability-related needs but are not directly involved in admission decisions. The least acceptable option is the Dean of Admissions and Office of Admissions staff. These options place the applicant's disability, apparent or non-apparent, directly in the view of the selection committee and could lead to implicit or explicit biases that may harm the applicant's chance of admission. Analysis of valid responses revealed that the majority (66.7%) utilized their admissions department in some regard—either the department staff or the Dean—as the primary point of contact, while only one-quarter (25.9%) adhered to best practices by designating the DRP. There was slight improvement when designating who communicates accommodation outcomes, with less than half (44.4%) of institutions involving their admissions offices and admissions deans. However, this progress should be viewed with caution, given the concurrent decline in institutions utilizing the DRP for finalizing requests (22.5%), resulting in the designation of OSA and other departments that lack expertise in disability accommodations. Additionally, the difference between the involvement of the Dean of Admissions and the general admissions office worsens during this final stage (communicating accommodations to the candidate). Specifically, only 7.4% of institutions assign the Dean of Admissions responsibility for the initial request, while 29.6% do so for the final contact phase. A similar pattern emerges when examining US and Canadian institutions separately, indicating considerable procedural oversights. Notably, compared to US respondents, Canadian institutions demonstrate a higher utilization rate of their DRP across both stages, yet rely more heavily on their admissions office to communicate accommodations to candidates. However, these differences in administrative structure did not translate into measurable disparities in applicant outcomes. It is possible that the relatively small sample of valid responses (n = 27) may have limited statistical power to detect subtle differences. However, this could also suggest that institutions may follow consistent internal policies that negate differences between designated administrative personnel.

### Documentation

The documentation requirements for interview day accommodations varied considerably across the 27 responding medical schools. While some institutions imposed specific requirements—such as a unique application form (18.5%), a clinician's letter (18.5%), or proof of past accommodations (14.8%)—a majority (55.6%) reported that no formal application or

documentation was necessary. This variation suggests a lack of standardized practices across medical schools and raises questions about transparency and accessibility in the accommodations process. The fact that over a quarter of schools (25.9%) selected "Other" and provided open-text responses further highlights the heterogeneity in institutional approaches and the need for clearer, more consistent guidelines.

### Technical standards

TS must be accessible for review by CWD; ideally TS are available on the admissions webpage. Over half (56.3%) of responding institutions confirmed posting their TS on their website, with the remainder leaving the question unanswered. Encouragingly, among those with posted TS, the majority (77.5%) confirmed placement on the admissions webpage. Impressively, over three-quarters of Canadian institutions (77.8%) posted TS on their website and 100% of those were on the admissions webpage.

Furthermore, institutions should guide applicants toward the TS early in the admissions process, enabling them to understand the program requirements and promptly assess their accommodation needs prior to acceptance. This is especially important given the significant financial, logistical, and emotional investment required to apply to medical school. Our analyses revealed that the majority (77.5%) of responding institutions followed this practice. As TS can be difficult to interpret and most candidates will not understand intricate clinical requirements within a medical education curriculum, applicants must be directed to a more knowledgeable source to help navigate their needs. Thus, it is concerning that less than half (43.7%) of institutions directed applicants to the Office of Disability within the TS.

### Limitations and future directions

The major limitation of our study was the 41.2% overall response rate, which raises concern for non-response bias. One such possibility is that institutions with more inclusive or established disability accommodation practices were more likely to respond, potentially skewing results toward more favorable findings. However, Canadian institutions achieved a response rate exceeding 50%. To encourage participation, not all questions were mandatory, and some institutions left questions unanswered. About half of respondents completed the survey, with the rest offering partial responses, leading to the omission of some questions due to insufficient sample sizes. Although the survey was piloted with admissions staff, the length of the survey potentially contributed to survey fatigue.

Future research should probe the areas of non-response to ascertain why these questions may have been left unanswered. To ensure better understanding of accommodation practices, institutional data tracking systems should be utilized through the DRP's office to ensure protected information is being collected on applicants and matriculants with disabilities. More research is needed on the accessibility of the medical school application process, especially the interview, given the shift to online formats post-COVID-19. While many argue that accommodations are unnecessary now that interviews have been moved to the remote format, this approach ignores the needs of many populations with disabilities, particularly those who are ASL-users who will require additional time and the presence of an interpreter in an interview to fully participate.

### Recommendations

Based on data collected in our study and previous literature, we present multiple recommendations for LCME-accredited medical institutions: (1) All medical schools should provide applicants with the procedure associated with requesting interview accommodations as early as possible. (2) To ensure fair treatment and minimize bias in the admissions process, medical institutions should designate the DRP as the initial contact for interview accommodation requests and to communicate the plan to the applicant. This ensures that knowledgeable individuals handle applicants' needs separate from the admissions process. (3) All medical schools should establish clear documentation requirements that are ideally consistent across institutions. (4) Institutions should (a) post TS on the admissions webpage, (b) direct applicants to review the TS

prior to the interview, and (c) include contact information for the Office of Disability within the TS. (5) Lastly, we encourage widespread adoption of the MMI format due to its demonstrated advantages in reducing bias and increasing reliability, alongside recommendation #1 to ensure all CWD can apply for accommodations as needed.

These recommendations are crucial for ensuring accessibility and support for all applicants, especially CWD, throughout the admissions process and beyond. As changes are implemented, we hope to see increased diversity amongst future physicians, which benefits the community by dispelling some of the inaccuracies and stigma surrounding persons with disabilities and fostering an increased understanding that could impact health outcomes.

## Supporting information

**S1 Table. Survey results by domain and full question text.** Summary of institutional responses by domain, detailing interview formats, applicant volumes, accommodation processes, and technical standards.
(TIF)

**S1 Appendix. Survey instructions.** Instructions provided to survey respondents outlining the purpose, scope, reference year (2018–2019 cycle), and support resources for survey completion, including IRB exemption details.
(DOCX)

**S2 Appendix. Survey questions.** Full text of the 2021 survey distributed to medical school admissions deans and disability resource providers, covering 2018–2019 admissions data, interview day accommodation procedures, current practices, and technical standards.
(DOCX)

**S1 File. Deidentified data.**
(XLSX)

## Acknowledgments

We would like to acknowledge the many medical students with disabilities who have shared their concerns over the lack of admissions accommodations with members of our team, as their experiences informed this study. We also would like to express appreciation to the medical school leaders who responded to our survey.

## Author contributions

**Conceptualization:** Aitan E. Magence, Fern R. Juster, Kristina H. Petersen.

**Data curation:** Sarah Leah Solomon, Kristina H. Petersen.

**Formal analysis:** Sarah Leah Solomon, Aitan E. Magence, Harry Haran, Kristina H. Petersen.

**Project administration:** Sarah Leah Solomon, Aitan E. Magence, Kristina H. Petersen.

**Supervision:** Kristina H. Petersen.

**Writing – original draft:** Sarah Leah Solomon, Kristina H. Petersen.

**Writing – review & editing:** Sarah Leah Solomon, Aitan E. Magence, Harry Haran, Fern R. Juster, Kristina H. Petersen.

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
