## [Decision Letter · Decision Letter 0]

15 Apr 2025

Dear Dr. Solomon,

Thank you for submitting your manuscript to PLOS ONE. After careful consideration, we feel that it has merit but does not fully meet PLOS ONE’s publication criteria as it currently stands. Therefore, we invite you to submit a revised version of the manuscript that addresses the points raised during the review process.

We look forward to receiving your revised manuscript.

Kind regards,

Vidya Ramkumar, Ph.D

Academic Editor

PLOS ONE

Journal Requirements:

4. We notice that your supplemental files [Supplemental Digital Content (SDC), Supplemental Digital Appendix 1 &2] are included in the manuscript file. Please remove them and upload them with the file type 'Supporting Information'. Please ensure that each Supporting Information file has a legend listed in the manuscript after the references list.

**Additional Editor Comments:**

This paper addresses an important aspect of disability inclusion at the level if academic institutions. The findings would be valuable, however the paper requires significant restructuring in reporting the survey and its analysis. The frameworks for reporting suggested by the reviewer may please be considered and a revised manuscript maybe submitted for consideration

Reviewers' comments:

Reviewer's Responses to Questions

**Comments to the Author**

1. Is the manuscript technically sound, and do the data support the conclusions?

Reviewer #1: Partly

2. Has the statistical analysis been performed appropriately and rigorously?

Reviewer #1: Yes

3. Have the authors made all data underlying the findings in their manuscript fully available?

Reviewer #1: Yes

4. Is the manuscript presented in an intelligible fashion and written in standard English?

Reviewer #1: Yes

Reviewer #1: Thank you for submitting your article related to disclosing disabilities during the medical school application process. The following comments are intended to help the authors improve their work.

Overall, this article could benefit from reading guidelines published by Artino and colleagues on reporting survey-based research. Specifically, Table 1 has a fantastic outline that will help guide the authors on how to structure the manuscript. The citation is:

Artino AR Jr, Durning SJ, Sklar DP. Guidelines for Reporting Survey-Based Research Submitted to Academic Medicine. Acad Med. 2018 Mar;93(3):337-340. doi: 10.1097/ACM.0000000000002094. PMID: 29485492.

Additionally, some of the statistics reported were not done in a clear manner. There are recommended ways to report statistical analyses from the American Psychological Association. This particular document does a good job explaining how to report your findings:

https://www.loraconnor.com/psych280/psych_280/toolbox-docs/Reporting%20Statistics%20in%20APA%20Format.pdf

Introduction

1. Line 78 introduces us to the term “candidates with disabilities” but there is no further defining characteristics for this. What types of disabilities are you talking about? Physical? Mental? You need to clarify these terms because they have multiple meanings.

2. Line 81: What is meant by “situational” interview?

3. Lines 91-93: The sentence beginning “In these cases, candidates may feel…” is really the problem statement you are attempting to address with this study. This particular sentence should be moved before this subsection and expanded upon to make note of the psychological and emotional turmoil candidates face when considering whether or not to disclose their disability during the candidacy process. It would then help the subsequent sections make more sense.

Methods

1. Beginning with Line 150, you refer to questions using numbers. You need to rewrite this to note categories of your survey. Readers should be able to get through your article without having to open up the survey to cross-reference. As it is now, I had to jump back and forth from the text to the survey to understand what you were discussing. It was very frustrating and if this was published I would stop reading the article.

2. Please review the Artino et al article about how to structure the Methods.

Results

1. For the tables, both in text and supplemental, I did not find any of them to be of value. Within the text you are summarizing the exact information in the table. What would be helpful is to have a table with the entire survey (question language included) with the data summarized. You could then reference a single table instead of having multiples with repeated data. This would also allow you to summarize main points of the descriptive data and point to important points of interest.

2. With regard to Tables 2 and 3, I am curious why some type of statistical analysis was not conducted with this information. It seems rather intriguing to look at the contact and designee for accommodation requests.

3. Lines 201-215: Again, see the APA document I referenced for more appropriate reporting of statistical analyses.

Discussion

• Lines 317-318: I am wondering if it is more important to have the accommodations policy present for applicants so they know the school will work with them regarding technical standards. If a candidate has not been accepted to the school, is it really important for the school to address accommodations specific to technical standards or is it okay to have a policy that addresses that? I could see an argument for both sides, but that is not how this is written.

• The limitations section could benefit from addressing non-response bias.

**Do you want your identity to be public for this peer review?** For information about this choice, including consent withdrawal, please see our Privacy Policy

Reviewer #1: **Yes: ** Gary L. Beck Dallaghan, Ph.D.

---

## [Author Response · Author response to Decision Letter 1]

28 May 2025

Title: Response to Reviewers

Manuscript Title: Disclosing Disabilities: Barriers for Medical School Applicants

Manuscript ID: PONE-D-24-47715

Authors: Sarah Solomon, Aitan E. Magence, Harry Haran, Fern R. Juster, and Kristina H. Petersen

Dear Dr. Vidya Ramkumar and Reviewers,

We sincerely thank you for your careful review of our manuscript and for the helpful comments and suggestions. We have addressed each point below and revised the manuscript accordingly. Below is our point-by-point response.

Response to Editorial-Specific Requirements:

Comment 1: Please ensure that your manuscript meets PLOS ONE's style requirements, including those for file naming.

Response: Thank you for providing the "Sample Main Body" reference document. We have revised the abstract and section headings to align with the required format. As our manuscript does not include figures, the remaining formatting notes related to figures were not applicable. Additionally, using the "Sample Title, Authors, Affiliations" document, we have completely revised the first page to conform to the required formatting. References and tables were reviewed to ensure compliance with the journal’s style guidelines, and the Supporting Information was extensively edited to meet the formatting requirements.

Comment 2: When completing the data availability statement of the submission form, you indicated that you will make your data available on acceptance. We strongly recommend all authors decide on a data sharing plan before acceptance, as the process can be lengthy and hold up publication timelines. Please note that, though access restrictions are acceptable now, your entire data will need to be made freely accessible if your manuscript is accepted for publication. This policy applies to all data except where public deposition would breach compliance with the protocol approved by your research ethics board. If you are unable to adhere to our open data policy, please kindly revise your statement to explain your reasoning and we will seek the editor's input on an exemption. Please be assured that, once you have provided your new statement, the assessment of your exemption will not hold up the peer review process.

Response: We are able to adhere to the data availability statement and are working with our team of authors to get that going.

Comment 3: Please include captions for your Supporting Information files at the end of your manuscript, and update any in-text citations to match accordingly.

Response: This has been updated in accordance with the journal requirements. Captions for the Supporting Information files have been included at the end of the manuscript, and all in-text citations have been updated accordingly.

Comment 4: We notice that your supplemental files [Supplemental Digital Content (SDC), Supplemental Digital Appendix 1 &2] are included in the manuscript file. Please remove them and upload them with the file type 'Supporting Information'. Please ensure that each Supporting Information file has a legend listed in the manuscript after the references list.

Response: This has been updated in accordance with the journal requirements.

Reviewer #1:

Comment 1: Overall, this article could benefit from reading guidelines published by Artino and colleagues on reporting survey-based research. Specifically, Table 1 has a fantastic outline that will help guide the authors on how to structure the manuscript. The citation is:

Artino AR Jr, Durning SJ, Sklar DP. Guidelines for Reporting Survey-Based Research Submitted to Academic Medicine. Acad Med. 2018 Mar;93(3):337-340. doi: 10.1097/ACM.0000000000002094. PMID: 29485492.

Additionally, some of the statistics reported were not done in a clear manner. There are recommended ways to report statistical analyses from the American Psychological Association. This particular document does a good job explaining how to report your findings:

https://www.loraconnor.com/psych280/psych_280/toolbox-docs/Reporting%20Statistics%20in%20APA%20Format.pdf

Response: Thank you for the comment and supplemental links, we have corrected our methods and statistical reporting based on the provided links.

Comment 2: Line 78 introduces us to the term “candidates with disabilities” but there is no further defining characteristics for this. What types of disabilities are you talking about? Physical? Mental? You need to clarify these terms because they have multiple meanings.

Response: Thank you for pointing this out. We agree that the term “candidates with disabilities” can encompass a wide range of conditions and should be clarified for the reader. In our revision, we clarified that our study focused on disabilities including physical, sensory, psychological, and learning disabilities, which encompasses both visible and invisible disabilities—such as chronic health conditions, mobility impairments, ADHD, and mental health disorders. This reflects our intention to capture the experiences of a broad spectrum of candidates who may require accommodations in the medical school admissions process.

Comment 3: Line 81: What is meant by “situational” interview?

Response: We revised the text to clarify that “situational interview” refers to interview formats that assess how applicants respond to hypothetical scenarios (such as Multiple Mini Interviews).

Comment 4: Lines 91-93 (Now 98-99 in the new version): The sentence beginning “In these cases, candidates may feel…” is really the problem statement you are attempting to address with this study. This particular sentence should be moved before this subsection and expanded upon to make note of the psychological and emotional turmoil candidates face when considering whether or not to disclose their disability during the candidacy process. It would then help the subsequent sections make more sense.

Response: We appreciate the reviewer’s suggestion; however, we chose to keep the sentence within the "Interview Day" subsection to preserve the paper’s chronological structure. The introduction overall transitions between interview logistics to technical standards, reflecting the applicant’s decision-making process. We believe the sentence is most effective here, as it captures the emotional complexity of deciding whether to disclose a disability—particularly when disabilities may become more visible or impactful during interviews. On that note, we have expanded this section to make the emotional turmoil associated with this decision more explicit.

Comment 5: Beginning with Line 150, you refer to questions using numbers. You need to rewrite this to note categories of your survey. Readers should be able to get through your article without having to open up the survey to cross-reference. As it is now, I had to jump back and forth from the text to the survey to understand what you were discussing. It was very frustrating and if this was published I would stop reading the article.

Response: Thank you for the comment. Based on this point we have reorganized our tables into a singular summary table of all questions analyzed in the discussion with a brief overview of the results. Additionally, based on Comment 7, we have used this single overview table to complement our prose.

Comment 6: Please review the Artino et al article about how to structure the Methods.

Response: Thank you for this comment. We appreciate the supplementary documents and reported our methods based on the provided information.

Comment 7: For the tables, both in text and supplemental, I did not find any of them to be of value. Within the text you are summarizing the exact information in the table. What would be helpful is to have a table with the entire survey (question language included) with the data summarized. You could then reference a single table instead of having multiples with repeated data. This would also allow you to summarize main points of the descriptive data and point to important points of interest.

Response: Please see the response to Comment 5.

Comment 8: With regard to Tables 2 and 3, I am curious why some type of statistical analysis was not conducted with this information. It seems rather intriguing to look at the contact and designee for accommodation requests.

Response: Thank you for this comment, initially we were unsure if this would be of any importance, due to space constraints we excluded the analysis portion of these tables but have included it in the revised manuscript.

Comment 9: Lines 201-215: Again, see the APA document I referenced for more appropriate reporting of statistical analyses.

Response: Thank you for this comment. We appreciate the supplementary documents and files and reported our analyses based on the provided information.

Comment 10: Lines 317-318 (Now 361-363 in the new version): I am wondering if it is more important to have the accommodations policy present for applicants so they know the school will work with them regarding technical standards. If a candidate has not been accepted to the school, is it really important for the school to address accommodations specific to technical standards or is it okay to have a policy that addresses that? I could see an argument for both sides, but that is not how this is written.

Response: We appreciate the reviewer’s thoughtful perspective. While we understand the rationale for addressing accommodations policy post-acceptance, we intentionally emphasized early access to technical standards (TS) and guidance on accommodations. Applicants with disabilities often face significant financial, logistical, and emotional burdens during the admissions process. Without early clarity on TS and available accommodations, candidates may invest substantial effort applying to programs that ultimately cannot—or will not—support their needs. From an equity standpoint, we believe institutions should support informed decision-making as early as possible, and thus maintain our emphasis on early TS accessibility and guidance. We have edited this paragraph in the discussion to more clearly reflect that concern.

Comment 11: The limitations section could benefit from addressing non-response bias.

Response: We agree that there is the potential for non-response bias, particularly given the possibility that institutions with more robust accommodation practices may have been more likely to respond. We have revised the limitations section to explicitly address this concern.

We appreciate the reviewers’ thoughtful feedback and hope that the revisions we have made have addressed the concerns raised. Please let us know if further clarification is needed.

Sincerely,

Dr. Sarah Solomon

on behalf of all authors

---

## [Decision Letter · Decision Letter 1]

6 Jun 2025

Disclosing disabilities: Barriers for medical school applicants

PONE-D-24-47715R1

Dear Dr. Solomon,

We’re pleased to inform you that your manuscript has been judged scientifically suitable for publication and will be formally accepted for publication once it meets all outstanding technical requirements.

Kind regards,

Vidya Ramkumar, Ph.D

Academic Editor

PLOS ONE

Additional Editor Comments (optional):

The revised version has restructured the survey and its analysis and adequately addressed all the comments provided.

Reviewers' comments:

Reviewer's Responses to Questions

**Comments to the Author**

Reviewer #1: All comments have been addressed

2. Is the manuscript technically sound, and do the data support the conclusions?

Reviewer #1: Yes

3. Has the statistical analysis been performed appropriately and rigorously?

Reviewer #1: Yes

4. Have the authors made all data underlying the findings in their manuscript fully available?

Reviewer #1: Yes

5. Is the manuscript presented in an intelligible fashion and written in standard English?

Reviewer #1: Yes

Reviewer #1: Thank you for revising your manuscript and addressing the concerns raised from the previous version. I am sufficiently satisfied with the work you have done and have no further suggestions for edits.

**Do you want your identity to be public for this peer review?** For information about this choice, including consent withdrawal, please see our Privacy Policy

Reviewer #1: **Yes: ** Gary L. Beck Dallaghan, Ph.D.

---

## [Editor Report · Acceptance letter]

PONE-D-24-47715R1

PLOS ONE

Dear Dr. Solomon,

I'm pleased to inform you that your manuscript has been deemed suitable for publication in PLOS ONE. Congratulations! Your manuscript is now being handed over to our production team.

Kind regards,

on behalf of

Dr. Vidya Ramkumar

Academic Editor

PLOS ONE